# Identification and Prioritization of Environmental Reproductive Hazards: A First Step in Establishing Environmental Perinatal Care

**DOI:** 10.3390/ijerph16030366

**Published:** 2019-01-28

**Authors:** Raphaëlle Teysseire, Patrick Brochard, Loïc Sentilhes, Fleur Delva

**Affiliations:** 1Department of Occupational Medicine, Bordeaux Hospital, 33076 Bordeaux, France; patrick.brochard@chu-bordeaux.fr (P.B.); fleur.delva@chu-bordeaux.fr (F.D.); 2Environmental health platform dedicated to reproduction, ARTEMIS center, 33076 Bordeaux, France; loic.sentilhes@chu-bordeaux.fr; 3Bordeaux Population Health Research Center, Inserm UMR1219-EPICENE, University of Bordeaux, 33000 Bordeaux, France; 4Department of Obstetrics and Gynecology, Bordeaux Hospital, 33076 Bordeaux, France

**Keywords:** reproductive health, perinatal care, preventative medicine, environmental exposure, occupational exposure, chemical hazards

## Abstract

In 2015, the International Federation of Gynecology and Obstetrics established the prevention of exposures to environmental reprotoxic substances as a priority for health professionals. However, available information about reproductive hazards is voluminous, dispersed, and complex, and this is a severe limitation for physicians to incorporate the prevention of environmental exposure into standard preventive care. One difficulty frequently cited by physicians is the lack of evidence-based information. The objective of our study was to identify a list of environmental chemical hazards to reproduction. We used lists present in relevant regulations or included in scientific reports or databases to identify reproductive hazards. The reproductive hazards were prioritized according to the strength of evidence concerning their impact on fertility or development of the offspring. We identified 1251 reproductive hazards. Our prioritization approach resulted in a high-priority classification for 462 risk factors belonging to the following eight classes: drugs (*n* = 206), metals (*n* = 116), pesticides (*n* = 38), organic solvents (*n* = 27), synthesizing and/or processing agents in industrial processes (*n* = 23), phthalates (*n* = 13), perfluorinated compounds (*n* = 13), and other compounds (*n* = 26). Despite the limitations of this work, the generated lists constitute a useful working basis to put in place innovative environmental preventive measures according to the principle of evidence-based medicine.

## 1. Introduction

A growing body of literature demonstrates the adverse effects of environmental exposure on reproductive health, including fertility, pregnancy, and postnatal development [1]. Occupational and non-occupational exposure to contaminants is now identified to be responsible for adverse outcomes. For example, exposure to alcohol, tobacco products, polychlorobiphenyls (PCBs), phthalates, radiation, or heat have been linked to decreased sperm concentration or semen quality [2]. Exposure to air-borne pollutants, lead, environmental tobacco smoke, pesticides, or organic solvents have been related to adverse birth outcomes [3,4]. There is consistent evidence for an association between exposure to heavy metals and polychlorinated biphenyls (PCBs) and the impairment of the postnatal neurodevelopment [5].

The accumulating evidence prompted the International Federation of Gynecology and Obstetrics (FIGO) to establish the prevention of exposures to reprotoxic substances as a priority for obstetricians, gynecologists, midwives, and the other perinatal health professionals in 2015. FIGO recommended that reproductive health providers learn about toxic chemicals and other harmful environmental exposures and take environmental histories during visits to educate patients about how to avoid toxic environmental chemicals [1].

Perinatal health providers do not generally ask patients about their environmental exposure except for alcohol, tobacco, and other drugs. Indeed, a national survey in the United States found that less than 20% of obstetricians routinely interview their patients about their environmental exposure [6]. These results are consistent with those of a French study, which found that few obstetricians, midwives, or general practitioners ask and advise women about their chemical exposure [7].

Although health professionals recognize the potential impact of the environment on reproductive health [8], they cite numerous barriers to counseling patients, one of which is the lack of evidence-based information about the risks [6,7]. Perinatal health professionals require science-based guidelines and resources about environmental reproductive hazards before they can develop such prevention-oriented actions [6]. 

Indeed, the available information is voluminous, dispersed, and complex, and its quality varies widely, depending on the data source. Reviews of the available epidemiological or toxicological data published in scientific journals can provide data on reproductive risk factors [2,3,4,5,9]. Some American and European regulations frame the classification of toxic chemicals to properly identify and communicate the hazards of substances, including reproductive impairments [10]. In addition, national programs and scientific agency publications tend to provide synthesized information about occupational and non-occupational hazards to human reproductive health [11,12,13,14].

The main objective of this study was to identify chemical hazards to reproductive health from various data sources. The secondary objective was to prioritize them according to the weight of the evidence. 

## 2. Methods

### 2.1. Identification of Chemical Reproductive Hazards

Environmental reproductive hazards include biological hazards, toxic chemicals, physical agents, and biomechanical or organizational demands that could harm sexual function, fertility, or in utero or postnatal development of the offspring. Here, we focused only on chemicals. Indeed, clinicians are generally familiar with the biological and physical risks. We also chose to exclude medical treatments that cannot be considered to be environmental exposures. Therefore, we included drugs when they could lead to occupational exposures in medical or research environments.

We identified environmental reproductive hazards using lists present in relevant regulations, scientific reports, or databases coming from recognized health agencies. All such classifications are based only on the intrinsic hazard of the substances and do not integrate the likelihood of exposure or the idea of risk.

We selected two regulatory texts: the European regulation on the classification, labeling, and packaging of substances and mixtures (CLP) [10] and Proposition 65, supervised by the Office of Environmental Health Hazard Assessment (OEHHA) of California [15].

#### 2.1.1. The CLP Regulation 

One of the main aims of the CLP regulation is to determine whether a substance or mixture displays properties that lead to a hazardous classification. The text distinguishes 10 hazard classes, including two especially relevant to evaluating reproductive hazards: The “germ cell mutagenicity” class, which is primarily concerned with substances that may cause mutations in the germ cells of humans that can be transmitted to their progeny.The “reproductive toxicity” class, which includes adverse effects on sexual function and fertility in adult males and females, as well as developmental toxicity in their offspring.

Within these two classes, chemicals are divided into three categories based on the standard of proof concerning their adverse health effects. 

Concerning “reproductive toxicity”, reliable evidence of an adverse effect in humans leads to a category 1A classification “known human reproductive toxicant”. Less rigorous data from human or animal studies result in a category 1B classification “presumed human reproductive toxicant”. Finally, insufficient data from human or animal studies to place the substance in category 1 result in a category 2 classification “suspected human reproductive toxicant”. We chose to select the three categories for our identification of reproductive risk factors.

Concerning “germ cell mutagenicity”, the classification in category 1A is based on positive epidemiological studies, whereas category 1B and 2 are principally based on positive in vivo mutagenicity studies or even in vitro studies. We chose to consider only categories 1A and 1B to guarantee a sufficient level of evidence.

Under the CLP Regulation, a substance must be classified according to harmonized classifications listed in Annex VI in order to ensure an adequate risk management in the European Union (EU). When a substance has no harmonized classification in Annex VI to the CLP but presents hazardous properties, it must be self-classified by the manufacturer or importer according to available information. In our work, we only considered the substances covered by the harmonized classification and labeling regulation registered in Annex VI to the CLP regulation in order to be as rigorous as possible concerning the scientific evidence. Hence, we excluded substances concerned by self-classification processes that were not subjected to a consensus [10].

#### 2.1.2. Proposition 65

Proposition 65, also known as the Safe Drinking Water and Toxic Enforcement Act of 1986, was created to inform Californians about their exposures to chemicals, and it established a list of substances known to cause adverse health effects, including cancers, birth defects, or other reproductive harm [15]. This classification incorporates chemicals identified by: 1) the American Labor Code, 2) two independent committees of scientific and health experts (the Carcinogen Identification Committee and the Developmental and Reproductive Toxicant Identification Committee), 3) authoritative bodies (the United States Environmental Protection Agency (U.S. EPA), U.S. Food and Drug Administration (U.S. FDA), National Institute for Occupational Safety and Health (NIOSH), the National Toxicology Program (NTP) of the US Department of Health and Human Services, and the International Agency for Research on Cancer (IARC)), and 4) agencies of the state or federal government. The list is reviewed at least annually. This classification does not prioritize the hazard depending on the scientific evidence [15].

We also incorporated four lists proposed by recognized health agencies or research institutions to complement these regulatory sources.

#### 2.1.3. The Monographs Published by the National Toxicology Program (NTP)

The NTP Center for the Evaluation of Risks to Human Reproduction (NTP-CERHR) was established by the NTP and the National Institute of Environmental Health Sciences (NIEHS), two U.S. agencies, in 1998 to address the impact of chemical exposure on human reproduction and development. In 2010, it became the NTP Office of Health Assessment and Translation (OHAT). The products of this approach are monographs on chemicals, including an expert panel report and public comments. The reports include information about the weight of evidence regarding each substance’s toxicity on laboratory animals and/or humans [12]. We chose to select the compounds with “clear evidence” of reproductive adverse effects. 

#### 2.1.4. The NIOSH List of Drugs

The NIOSH of the Centers for Disease Control and Prevention (CDC) has made a list of hazardous drugs for health professionals, including those that affect reproductive health. The first list, created in 2004, has been regularly updated (last version in 2016). For each drug, the potential to cause a birth defect is indicated using the five FDA pregnancy risk categories, established prior to 2015 (A, B, C, D, or X). Three of these categories indicate a risk to reproductive health: category C includes drugs for which animal studies have shown an adverse effect on fetuses; category D incorporates drugs for which there is positive evidence of human fetal risk; and category X includes substances for which studies in animals or humans have demonstrated abnormalities and/or a human fetal risk [11].

#### 2.1.5. The DEMETER Database

The DEMETER (Documents pour l’Évaluation Médicale des produits Toxiques vis-à-vis de la Reproduction) database was developed by the French National Research and Safety Institute for the Prevention of Occupational Accidents and Diseases (INRS) to summarize information about reproductive hazards. The database incorporates information sheets written by toxicology experts based on bibliographic studies. Human and animal data are differentiated. The weight of evidence concerning adverse outcomes on reproductive health is specified for each chemical [13]. We only considered substances with limited or sufficient evidence from studies in humans and/or experimental animals.

#### 2.1.6. The Priority List of Chemicals Developed within the EU-Strategy for Endocrine Disrupters (EDs)

In 1999, the Commission adopted the ‘Community strategy for Eds’, which included the establishment of a priority list of substances. Three reports were published between 2000 and 2007 [14,16,17].

Each chemical has been assigned to a category related to the strength of evidence for endocrine disruption: category 1—evidence of endocrine disrupting activity in at least one species using intact animals, category 2—at least some in vitro evidence of biological activity related to endocrine disruption, and category 3—no evidence of endocrine disrupting activity or no data available [14]. We considered only category 1 and selected the substances found on the other selected lists, as these lists are not specific for reproductive hazards. 

### 2.2. Prioritization of Environmental Reproductive Hazards

The main objective for identifying reproductive hazards was to help health professionals assess their patients’ environments to propose appropriate and proportionate prevention-oriented measures. We chose a pragmatic approach by prioritizing the substances according to the strength of evidence concerning their reproductive hazards. Our prioritization approach included three categories based on the quality of the studies used to evaluate the reprotoxic compounds.
Category 1 included high-priority chemical risk factors for which adverse health effects have been demonstrated in epidemiological studies or documented in robust animal studies.Category 2 included the medium-priority chemical risk factors for which adverse reproductive outcomes have been observed in human or animal studies (but with results not robust enough to rise to a category 1 classification) and appeared in more than one classification.Category 3 contained low-priority chemical risk factors for which adverse reproductive outcomes have been observed in human or animal studies or in vitro experiments but appeared in only one classification.

As mentioned previously, some selected classifications already included a prioritization of the substances’ intrinsic toxicities (lists from the CLP regulation, the NTP-OHAT monographs, DEMETER data sheets, and the NIOSH report). Some other classifications did not do so by default (lists from Proposition 65 and EDs). We considered the weight of evidence for this last category to be low.
Category 1 included the following:
○Substances or mixtures classified by the CLP Regulation as reproductive toxicants in categories R1A and R1B.○Compounds identified in the DEMETER database for which there was sufficient or limited evidence of adverse health effects in epidemiological studies or sufficient evidence documented in robust animal studies.○Toxicants studied by the NTP-OHAT with clear evidence of adverse developmental or reproductive effects in humans or laboratory animals.○Substances listed in the report produced by the NIOSH.Categories 2 and 3 included the following:
○Substances or mixtures classified by the CLP Regulation as reproductive toxicants in category R2 or as a mutagen in categories 1A or 1B.○Toxicants registered on the OEHHA list.○Compounds identified in the DEMETER sheets for which there is sufficient or limited evidence of adverse health effects in animal studies.○Toxicants studied by the NTP-OHAT with limited evidence of adverse developmental or reproductive effects in laboratory animals.○Chemicals ranked in category 1 on the priority list of chemicals developed within the EU-Strategy for ED.

We then performed a search and merged duplicates based on the Chemical Abstracts Service (CAS) registry number and the name of the substance. The chemical families listed in some of our sources have been broken down into individualized substances recorded in other lists when necessary.

Finally, we removed the EDs that were not included in lists other than that developed within the EU-Strategy for EDs.

Figure 1 summarizes our method for identifying and prioritizing reproductive risks factors from existing classifications.

## 3. Results

### 3.1. Results of the Identification

The extraction of the lists of reproductive hazards from the identified classifications was performed on December 11, 2017. From the list provided by the CLP regulation, we identified 27 substances or mixtures classified in category R1A, 230 classified in category R1B, 143 classified in category R2, 0 classified in category M1A, and 427 classified in category M1B. From the Proposition 65 list, we extracted 313 chemicals that could cause adverse developmental outcomes or female or male fertility disorders. Among the monographs published by the NTP-OHAT, we identified 12 chemicals with clear evidence of adverse effects on laboratory animals, two substances with limited evidence of adverse effects on laboratory animals, and two substances with clear evidence of no adverse effects. From the list established by the NIOSH, we obtained 216 drugs for which studies in animals or humans have demonstrated an adverse effect on the fetus. From the data sheets of the DEMETER database, we obtained 38 substances with limited or sufficient evidence from studies in humans and/or with sufficient evidence from studies in experimental animals and 44 substances with limited evidence from studies in experimental animals. From the priority list of chemicals developed within the EU-Strategy EDs, we obtained 194 substances. 

### 3.2. Results of the Prioritization

Table 1 shows the results of the identification exercise after matching and merging according to the CAS number and/or the substance’s name. In some lists, we could observe discrepancies between the number of substances identified during the first step of our method and the number of chemical identified after matching and merging. For instance, we previously identified 313 chemicals from the Proposition 65 list, whereas we counted 333 substances after the procedure. Proposition 65 listed polybrominated biphenyls and polychlorinated biphenyls without detailing the congeners belonging to this family, whereas the list of EDs recorded these chemicals individually. In our method, we considered that all these substances registered on the EDs’ list were also listed in Proposition 65. Therefore, the number of substances identified from Proposition 65 increased after the matching and merging process. Finally, we observed a total overlap across the sources for 256 substances (e.g., difference from sum of columns and the line “total of chemicals”).

Table 1 shows the results of the prioritization procedure. As established in our method, we removed the EDs that did not appear in the classifications other than that generated by the European Commission. Finally, we classified 462 substances in Category 1, 79 chemicals in Category 2, and 710 compounds in Category 3. We observed some overlaps across sources within each category. Proportionally, overlaps were more important in Category 2 than in Category 1 and were null in Category 3. This is directly linked to our rules of prioritization. Thus, in Category 1, all the 98 substances extracted from Proposition 65 and the 15 chemicals from the EU list of EDs were recorded in other classifications as well. Almost all the substances studied by the NTP-OHAT were also identified as harmful by the CLP regulation or the DEMETER Database.

Table 2 shows all the chemicals identified as Category 1 “high-priority chemical risk factors”. We divided the compounds by eight classes to facilitate the analysis of the results: drugs, metals and metalloids, pesticides, organic solvents, synthesizing and/or processing agents in industrial processes, phthalates, perfluorinated compounds, and other compounds. Although only chemicals from Category 1 have been presented in Table 2, Appendix A including the full list of hazards from all the categories with their CAS numbers is available on request. For each compound, the lists on which a substance has been registered are specified.

## 4. Discussion

### 4.1. Main Findings

Our method allowed us to identify 1251 environmental reproductive hazards from two regulatory lists and four health agency scientific reports or databases. Our prioritization approach resulted in a high-priority classification for 462 risk factors. These substances belong to the following eight classes of chemicals: drugs (*n* = 206), metals and metalloids (*n* = 116), pesticides (*n* = 38), organic solvents (*n* = 27), synthesizing and/or processing agents in industrial processes (*n* = 23), phthalates (*n* = 13), perfluorinated compounds (*n* = 13), and other compounds (*n* = 26).

### 4.2. Strengths and Weaknesses

Our method enabled us to establish a relatively exhaustive list of risk factors, more complete than the classifications considered separately. Notably, we identified more reproductive hazards than those registered on the regulatory lists. Indeed, the classifications we used were established for various reasons and therefore cover various fields. For example, the CLP regulation applies to substances and mixtures supplied to the community but excludes drugs, which are included in the Proposition 65 and NIOSH lists.

We identified a large number of reproductive hazards (*n* = 1251). One strength of our method was the prioritization of the risk factors, based on the weight of the available scientific evidence, to determine their negative effects on reproductive health. Indeed, it is essential that clinical practice is based on the available scientific data to propose adequate and risk-proportionate preventive measures.

The principal limitation of our study was that the classifications we used do not necessarily reflect our clinical objective. They often omit complex mixtures, substances unintentionally generated by industrial processes, and/or prohibited chemicals. However, in the latter case, these substances can be relevant to our approach, because they persist in the environment and can lead to exposure.

In addition, some risk factors cannot be registered, despite scientific evidence of an adverse impact on reproduction in the scientific literature because of the time that is required for scientific expertise, although the lists are generally updated on a regular basis.

These two limitations can result in the non-identification of relevant risk factors or an unfavorable ranking of the substances in a lower category than they should be. For example, tobacco smoke is only registered on the Proposition 65 list and is ranked on our list as a Category 3 low-priority risk factor. However, there is sufficient published data to confirm the deleterious effects, which promote adverse pregnancy outcomes in humans [3,4,18]. Consequently, we should prioritize tobacco smoke as a high-priority risk factor (Category 1).

These lists should thus be completed by a bibliographic or expert review. Positive scientific evidence in publications should allow for the identification of new risk factors or reclassification to a higher priority level. This approach implies a chemical-by-chemical evaluation, such as that developed in other fields by independent organizations, such as the International Agency for Research on Cancer (IARC). Thus, one of the IARC’s missions is to classify agents based on their potential carcinogenicity according to the available evidence to guide cancer prevention.

Another example is given by the “Navigation Guide Systematic Review Methodology”, a systematic and transparent review that has been developed in the environmental health field to aid government agencies, professional societies, and healthcare organizations in improving the outcome of patients and, ultimately, the general health of the population [19]. However, this approach is time consuming and requires human resources that are not currently available.

Another possibility would be to incorporate more existing classifications. We cannot exclude the existence of additional scientific studies, especially national studies that may have not been included in this study. However, we chose to exclude substances or mixtures not registered in Annex VI to the CLP that nevertheless concerned by self-classification. Indeed, we based our analysis on validated and harmonized data to ensure adequate risk management. However, we could have included some of these chemicals, hence assigning them a low level of evidence. However, according to the Classification and Labelling Inventory database available on the ECHA website, a total of 4699 substances are classified as reprotoxic (R1A, R1B, or R2). There are also 924 substances classified as a mutagen (M1A, M1B, or M2). This information can be taken into account on a case-by-case basis in clinical practice by consulting the ECHA website [20].

### 4.3. Findings in Relation to Other Studies

Much of the scientific literature on the identification of reproductive risk factors is composed of reviews of epidemiological studies and/or toxicological data [2,3,4,5]. These analyses often concern a limited number of risk factors that partially match those on our list. We found two studies, which identified chemicals with reprotoxic properties, prioritized on the basis of the estimated exposure of the general [21] or worker population [22].

Some classes of chemicals of the highest concern are also included in our list of high-priority risk factors, such as phthalates, metals, alkylphenols, and glycol ethers [21]. Hence, practitioners should be vigilant concerning patient exposure to these risk factors in clinical practice.

However, other chemicals were not included (such as drugs) or not classed according to the same priority ranking. These differences can be explained by various factors, including the use of different scientific sources for the identification of the risk factors, a possible update of the lists in common, and divergence in the prioritization approaches. Indeed, we prioritized the substances based on the weight of the evidence of reprotoxic effects, whereas the other studies calculated a score based on exposure.

### 4.4. Implications and Perspectives

Our method was developed to identify and prioritize environmental reproductive hazards according to science-based information. Thus, we now have a list of risk factors of interest, ranked by current scientific concern. However, these lists cannot be directly used by perinatal health professionals in assessing patient exposure to reproductive factors. Nevertheless, this exercise represents a first step in establishing innovative environmental perinatal care. Indeed, assessment of the risk on reproductive health and the application of proportional preventive measures will require crossing information concerning the hazard related to a substance with the cumulative level of occupational and/or non-occupational exposure. This demands specialized skills, and it may be difficult for health professionals to link such risks with environmental exposure circumstances by themselves. Indeed, several studies in France and other countries have reported difficulties for perinatal health professionals to properly inform their patients in their regular practice about environmental risks. A lack of training and knowledge in environmental health, the lack of evidence-based information, and the short duration of consultations were the most frequently cited difficulties [6,7].

These elements favor the development of dedicated tools and medical structures to assist health professionals in this environmental health approach. One example is the Bordeaux University Medical Center, which has a specialized center dedicated to the evaluation of environmental risks on reproductive health, called the “Centre ARTEMIS” [23]. In this context, a specialized interview guide has been developed in order to detect the environmental exposure circumstances to reproductive hazards during consultations. Then, several prevention-oriented measures are proposed, depending on the priority category of the risk factor detected and the level of exposure: elimination of the risk factor from the patient’s environment, mitigation of the risk by proposing collective and individual protective equipment in an occupational setting, and changing attitudes concerning household practices (reasoned choice of products, aeration, etc.). At the same time, reproductive health professionals receive information concerning patient exposure, allowing the incorporation of environmental health in global medical care and progressive training of the perinatal health providers.

## 5. Conclusions

Our method allowed us to identify 1251 environmental reproductive hazards from existing classifications. Our prioritization approach resulted in the high-priority classification of 462 reproductive risk factors, grouped in eight classes. Exposure to these families should be given priority by perinatal health providers during patient interviews. We also identified 79 reproductive hazards with a medium-priority classification and 710 substances or mixtures with a low-priority ranking. It would be informative to complete this work with a bibliographic or expert review to address the inherent limits of our methodology and add robustness to the study. Nonetheless, the generated lists constitute a useful working basis to put in place innovative environmental preventive measures according to the principle evidence-based medicine.

## Figures and Tables

**Figure 1 ijerph-16-00366-f001:**
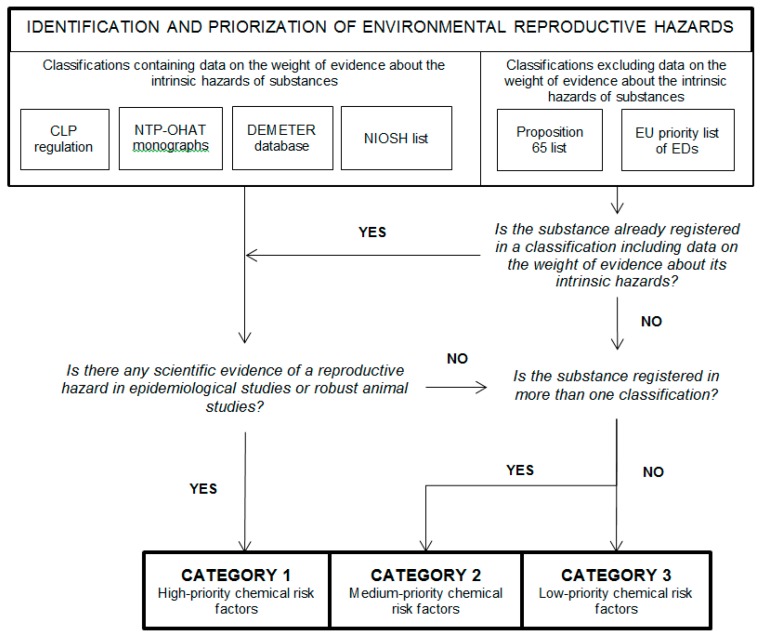
Method for identifying and prioritizing risk factors. CLP: the European regulation on the classification, labeling, and packaging of substances and mixtures; NTP-OHAT: National Toxicology Program Office of Health Assessment and Translation; DEMETER: Documents pour l’Évaluation Médicale des produits Toxiques vis-à-vis de la Reproduction; NIOSH: National Institute for Occupational Safety and Health; and ED: Endocrine Disrupters.

**Table 1 ijerph-16-00366-t001:** Number of environmental reproductive hazards registered on each classification, prioritized according to our method.

Lists		Chemicals Identified after Matching and Merging	Category 1 High-Priority Chemical Risk Factors	Category 2 Medium-Priority Chemical Risk Factors	Category 3 Low-Priority Chemical Risk Factors
Regulatory lists	The CLP Regulation List of the reprotoxic substances or mixtures registered in Annex VI	378	246	31	101
	The CLP Regulation List of the mutagenic substances or mixtures registered in Annex VI	427	15	8	404
	Proposition 65 list	333	98	57	178
Lists established by health agencies or research organizations	Monographs published by the NTP-OHAT	12	10	0	2
	The NIOSH List of Antineoplastic and Other Hazardous Drugs in Healthcare Settings	205	205	0	0
	Priority list of chemicals developed within the EU-Strategy for Endocrine Disrupters	70	15	55	0
	DEMETER database	82	46	11	25
Total of chemicals		1251	462	79	710

CLP: the European regulation on the classification, labeling, and packaging of substances and mixtures; NTP-OHAT: National Toxicology Program Office of Health Assessment and Translation; DEMETER: NIOSH: National Institute for Occupational Safety and Health.

**Table 2 ijerph-16-00366-t002:** High-priority chemical risk factors.

Classes	Chemicals
Drugs (*n* = 206)	Antineoplastic agents (*n* = 95) ado-trastuzumab emtansine; altretamine; anastrozole; arsenic trioxide; axitinib; azacitidine; belinostat; bendamustine; bexarotene; bicalutimide; bleomycin; bortezomib; bosutinib; brentuximab vedotin; busulfan; cabazitaxel; cabozantinib; capecitabine; carboplatin; carfilzomib; carmustine; chlorambucil; cisplatin; cladribine; clofarabine; crizotinib; cytarabine; dabrafenib; dacarbazine; dactinomycin; dasatinib;decitabine;degarelix; docetaxel; droloxifen; enzalutamide; epirubicin; eribulin; erlotinib; estramustine; etoposide; everolimus; exemestane; floxuridine; fludarabine; fluorouracil; flutamide; fulvestrant; gemcitabine; gemtuzumab ozogamicin; hydroxyurea; idarubicin; ifosfamide; imatinib; irinotecan; ixabepilone; letrozole; lomustine; mechlorethamine; melphalan; mercaptopurine; methotrexate; mitomycin; mitotane; nelarabine; nilotinib; omacetaxin; oxaliplatin; paclitaxel; pazopanib; pemetrexed; pentostatin; pertuzumab; pomalidomide; ponatinib; pralatrexate; regorafenib; romidepsin; sorafenib; sunitinib; temozolomide; temsirolimus; teniposide; thioguanine; topotecan; toremifene; trametinib; triptorelin; valrubicin; vandetanib; vemurafenib; vinorelbine; vismodegib; vorinostat; ziv-aflibercept Other drugs (*n* = 111) abacavir; abiraterone acetate; acitretin; afatinib; alitretinoin; ambrisentan; apomorphine; azathioprine; bosentan; carbamazepine; cetrorelix; chloramphenicol; choriogonadotropin; cidofovir; clomiphene; clorazepate dipotassium; colchicine; cyclophosphamide (anhydrous); cyclosporine; daunorubicin hydrochloride; deferiprone; dexrazoxane; diethylstilbestrol (des); dinoprostone; divalproex; doxorubicin hydrochloride (adriamycin); dronedarone; dutasteride; entecavir; ergonovine/methylergo-novine; eslicarbazepine; estradiol estrogen/progester-one combinations; estrogens, esterified; estrogens, conju-gated; estropipate; finasteride; fingolimod; fluconazole; fluoxymesterone; fosphenytoin; ganciclovir; ganirelix; gonadotropin, chorionic; goserelin; histrelin acetate; icatibant; leflunomide; lenalidomide; leuprolide acetate; liraglutide; lomitapide; macitentan; medroxyprogesterone acetate; megestrol acetate; menotropins; methimazole; methyltestosterone; mifepristone; mipomersen; misoprostol; mitoxantrone hydrochloride; mycophenolate mofetil; mycophenolic acid; nafarelin acetate; nitrous oxide; ospemifene; oxcarbazepine; oxytocin; palifermin; paliperidone; pamidronate; paroxetine; pasireotide; peginesatide; pentetate calcium triso-dium; phenoxybenzamine; phenytoin; pipobroman; plerixafor; procarbazine hydrochloride; progestins; raloxifene; rasagiline; ribavirin; riociguat; risperidone; sirolimus; spironolactone; streptozocin (streptozotocin); tacrolimus; tamoxifen citrate; telavancin; temazepam; teriflunomide; testosterone; thalidomide; tofacitinib; topiramate; tretinoin; ulipristal; uracil mustard; valganciclovir; valproate (valproic acid); vigabatrin; vinblastine sulfate; vincristine sulfate; voriconazole; warfarin; zidovudine; ziprasidone; zoledronic acid; zonisamide
Metals and metalloids (*n* = 116)	Lead and its compounds (*n* = 15) lead hexafluorosilicate; silicic acid, lead nickel salt; lead compounds with the exception of those specified elsewhere in the CLP regulation; lead alkyls; lead diazide, lead azide; lead chromate; lead di(acetate); trilead bis(orthophosphate); lead acetate, basic; lead(II) methanesulphonate; lead sulfochromate yellow; lead chromate molybdate sulfate red; lead hydrogen arsenate; lead; lead styphnate Nickel and its compounds (*n* = 69) tetracarbonylnickel; nickel dihydroxide; nickel hydroxide; nickel sulfate; [carbonato(2-)]tetrahydroxytrinickel[µ-[carbonato(2-)-O:O’]] dihydroxy trinickel; carbonic acid, nickel salt; nickel carbonate; nickel dichloride; nitric acid, nickel salt; nickel dinitrate; slimes and sludges, copper electrolytic refining, decopperised, nickel sulfate; nickel diperchlorate; diammonium nickel bis(sulfate); nickel dipotassium bis(sulfate); nickel bis(sulfamidate); nickel bis(tetrafluoroborate); formic acid, copper nickel salt; formic acid, nickel salt; nickel diformate; nickel acetate; nickel di(acetate); nickel dibenzoate; nickel bis(4-cyclohexylbutyrate); nickel(II) stearate; nickel dilactate; nickel(II) octanoate; nickel dibromide; nickel diiodide; nickel difluoride; nickel potassium fluoride; nickel hexafluorosilicate; nickel selenite; nickel dithiocyanate; nickel dichromate; nickel dichlorate; ethyl hydrogen sulfate, nickel(II) salt; nickel dibromate; dimethylhexanoic acid nickel salt; (isodecanoato-O)(isononanoato-O)nickel; (isooctanoato-O)(neodecanoato-O)nickel; nickel bis(isononanoate); (2-ethylhexanoato-O)(isodecanoato-O)nickel; nickel(II) palmitate; nickel 3,5-bis(tert-butyl)-4-hydroxybenzoate (1:2); nickel(II) neononanoate; bis(D-gluconato-O1,O2)nickel; fatty acids, C8-18 and C18-unsatd., nickel salts; nickel(II) neoundecanoate; nickel(II) propionate;2-ethylhexanoic acid, nickel salt; nickel bis(benzenesulfonate); citric acid, ammonium nickel salt; nickel(II) neodecanoate; nickel(II) isodecanoate; (isononanoato-O)(neodecanoato-O)nickel; nickel(II) isooctanoate; (isononanoato-O)(isooctanoato-O)nickel; (2-ethylhexanoato-O)(isononanoato-O)nickel; nickel bis(2-ethylhexanoate); (isodecanoato-O)(isooctanoato-O)nickel; fatty acids, C6-19-branched, nickel salts; nickel(II) trifluoroacetate; citric acid, nickel salt; (2-ethylhexanoato-O)(neodecanoato-O)nickel; neodecanoic acid, nickel salt; nickel(II) hydrogen citrate; nickel isooctanoate; 2,7-naphthalenedisulfonic acid, nickel(II) salt Boron and its compounds (*n* = 13) perboric acid (H3BO2(O2)), monosodium salt, trihydrate; perboric acid, sodium salt, monohydrate, perboric acid, sodium salt; disodium octaborate tetrahydrate, disodium octaborate anhydrous; dibutyltin hydrogen borate; boric acid; diboron trioxide; tetraboron disodium heptaoxide, hydrate; orthoboric acid, sodium salt; disodium tetraborate decahydrate; sodium peroxometaborate; di-sodium tetraborate anhydrous; boric acid; sodium perborate Other compounds (*n* = 19) slimes and sludges, copper electrolyte refining, decopperised; potassium dichromate; ammonium dichromate; sodium dichromate; sodium chromate; cobalt dichloride; cobalt sulfate; cobalt di(acetate); cobalt dinitrate; cobalt carbonate; gallium arsenide; cadmium fluoride; cadmium chloride; cadmium sulphate; tributyltin compounds; dibutyltin dichloride; 2-ethylhexyl 10-ethyl-4,4-dioctyl-7-oxo-8-oxa-3,5-dithia-4-stannatetradecanoate; dibutyltin dilaurate; mercury and mercury compounds
Pesticides (*n* = 38)	1,2-dibromo-3-chloropropane; azafenidin; benomyl; binapacryl; brodifacoum; bromadiolone; bromomethane; carbendazim; carbetamide; chlorophacinone; coumatetralyl; cycloheximide; cyproconazole; difenacoum; difethialone; dinocap; dinoseb; dinoterb; epoxiconazole; etacelasil; flocoumafen; fluazifop-butyl; flumioxazin; flusilazole; glufosinate ammonium; imidazole; ketoconazole; linuron; nitrofen; quizalofop-p-tefuryl; salts and esters of dinoseb; salts and esters of dinoterb; silafluofen; thiacloprid; triadimenol; tridemorph; triflumizole; vinclozolin
Organic solvents (*n* = 27)	Glycol ether family (*n* = 11) 1,2-diethoxyethane; 2-methoxyethanol; 2-ethoxyethanol; 1,2-dimethoxyethane;2-methoxypropanol; bis(2-methoxyethyl) ether; 1,2-bis(2-methoxyethoxy)ethane; 2-methoxyethyl acetate; 2-ethoxyethyl acetate; 2-methoxypropyl acetate; methoxyacetic acid 1,2,3-trichloropropane Other solvents (*n* = 16) 2,3-epoxypropan-1-ol; 2-butanone; dimethylacetamide; dimethylformamide; ethanol; ethylene glycol; formamide; Hexan-2-one; methanol; methylacetamide; methylformamide; N-methylpyrrolidone; Tetrahydro-2-furylmethanol; toluene; xylenes
Synthesizing and/or processing agent in industrial processes (*n* = 23)	1,3-diphenylguanidine; 1-bromopropane; 2-(2-aminoethylamino)ethanol; 2,3-epoxypropyl methacrylate; 2-bromopropane; 2-methyl-1-(4-methylthiophenyl)-2-morpholinopropan-1-one; acrylamide; carbon disulphide; chloroform; diphenylether; octabromo derivate; ethylene thiourea; methyl isocyanate; N-ethyl-2-pyrrolidone; nitrobenzene; phenol, (tetrapropenyl) derivatives; phenol, 2-dodecyl-, branched; phenol, 3-dodecyl-, branched; phenol, 4-dodecyl-, branched; phenol, dodecyl-, branched; p-phenylenediamine; R-glycidol; thiourea; trixylyl phosphate
Phthalates (*n* = 13)	1,2-benzenedicarboxylic acid; 1,2-benzenedicarboxylic acid, dipentylester, branched and linear; benzyl butyl phtalate; bis(2-ethylhexyl) phthalate; bis(2-methoxyethyl) phthalate; dibutyl phthalate; dicyclohexyl phthalate; dihexyl phthalate; diisobutyl phthalate; diisopentylphthalate; di-n-butyl phthalate (DBP); di-n-pentyl phthalate; n-pentyl-isopentylphthalate
Perfluorinated compounds (*n* = 13)	ammonium nonadecafluorodecanoate; ammonium perfluorooctane sulfonate; ammoniumpentadeca- fluorooctanoate; diethanolamine perfluorooctane sulfonate; lithium perfluorooctane sulfonate; nonadecafluorodecanoic acid; perfluorononan-1-oic acid; perfluorononan-1-oic acid ammonium salts; perfluorononan-1-oic acid sodium salts; perfluorooctane sulfonic acid; perfluorooctanoic acid; potassium perfluorooctanesulfonate; sodium nonadecafluorodecanoate
Other compounds (*n* = 26)	Bisphenols (*n* = 2) bisphenol A; 4,4-isobutylethylidenediphenol Chelating agent (*n* = 2) quinolin-8-ol; EDTA Oxides of carbon (*n* = 1) carbon monoxide Hydrocarbons (*n* = 2) benzo[a]pyrene; pitch, coal tar, high-temp. Other compounds (*n* = 19) (dimethylamino)thioacetamide hydrochloride[6,9-dihydro-9-[[2-hydroxy-1-(hydroxymethyl)ethoxy]methyl]-6-oxo-1H-purin-2-yl]acetamide; 1,2-benzenedicarboxylic acid, dihexyl ester, branched and linear; 2-[2-hydroxy-3-(2-chlorophenyl)carbamoyl-1-naphthylazo]-7-[2-hydroxy-3-(3-methylphenyl)carbamoyl-1-naphthylazo]fluoren-9-one; 2-butyryl-3-hydroxy-5-thiocyclohexan-3-yl-cyclohex-2-en-1-one; 2-ethylhexyl[[[3,5-bis(1,1-dimethylethyl)-4-hydroxyphenyl]methyl]thio]acetate; 2-ethylhexyl-2-ethylhexanoate; 3-ethyl-2-methyl-2-(3-methylbutyl)-1,3-oxazolidine; 4-tert-butylbenzoic acid; 7-methoxy-6-(3-morpholin-4-yl-propoxy)-3H-quinazolin-4-one; chloro-N,N-dimethylformiminium chloride; cyclic 3-(1,2-ethanediylacetale)-estra-5(10),9(11)-diene-3,17-dione; methyl-ONN-azoxymethyl acetate; potassium 1-methyl-3-morpholinocarbonyl-4-[3-(1-methyl-3-morpholinocarbonyl-5-oxo-2-pyrazolin-4-ylidene)-1-propenyl]pyrazole-5-olate reaction mass of: 1,3,5-tris(3-aminomethylphenyl)-1,3,5-(1H,3H,5H)-triazine-2,4,6-trione; reaction mass of: 4-[[bis-(4-fluorophenyl)methylsilyl]methyl]-4H-1,2,4-triazole; reaction mass of: disodium 4-(3-ethoxycarbonyl-4-(5-(3-ethoxycarbonyl-5-hydroxy-1-(4-sulfonatophenyl)pyrazol-4-yl)penta-2,4-dienylidene)-4,5-dihydro-5-oxopyrazol-1-yl)benzenesulfonate; tetrahydrothiopyran-3-carboxaldehyde; tris(2-chloroethyl)phosphate

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
