# Peer review of "Identification and Prioritization of Environmental Reproductive Hazards: A First Step in Establishing Environmental Perinatal Care"

_ijerph, 2019, doi:10.3390/ijerph16030366_

Reviewer 1 Report

Focusing on environmental reproductive hazards, this paper integrated several existing lists. The author identified and prioritized a list of chemical hazards, which may be used by physicians in perinatal preventive-care. This list could also be useful for industrial hygienists to identify reproductive risk factors. For example, one possible practice of using the list is to develop software or job exposure matrix (JEM) based on job modules to link the occupational/environmental activities to the identified chemicals, and thus quickly identify and evaluate hazard risks based on the intrinsic hazard of chemicals.

The reviewer has the following comments:

1. 4.1 Main findings shall be moved to section 3. Results.

2. In Table 2 "Other compounds", there shall be n=26.

3. On page 10, this is the same issue of number of "other compounds". Numbers of “4.1 Main findings” do not sum up to 462 factors. Also, the author shall double check the numbers in the abstract. 

4. On page 12, there is a typo: "Aknowledgment" shall be “Acknowledgement”.

5. Suggest adding 462 category 1 chemicals in the supplemental files, even though Category 1 chemicals were reported in table 2. Adding CAS number would help to develop software/interview modules to overcome the difficulty for professionals to link such risks with environmental/occupational circumstances.

Author Response

We thank the reviewer for his valuable comments and suggestions. We have made changes to the article in an effort to make it more intelligible and comprehensive. Hereinafter, we give our point-by-point answers to the concerns raised.

The reviewer made the five following comments:

1)    4.1 Main findings shall be moved to section 3. Results.

Summarizing our main findings at the beginning of the discussion seemed important in respect to providing a clear and accessible understanding of our results. Nonetheless, it is true that the article could gain in clarity were the main classes of chemicals presented in section 3. Results. Consequently, we made the following change to the description of Table 2 (line 200):

We divided the compounds by class to facilitate the analysis of the results. Table 2 shows all the chemicals identified as Category 1 “high-priority chemical risk factors”.

“Table 2 shows all the chemicals identified as Category 1 “high-priority chemical risk factors”. We divided the compounds by eight class to facilitate the analysis of the results: drugs, metals and metalloids, pesticides, organic solvents, synthesizing and/or processing agents in industrial processes, phthalates, perfluorinated compounds and other compounds.”

2)    In Table 2 "Other compounds", there shall be n=26.

3)    On page 10, this is the same issue of number of "other compounds". Numbers of “4.1 Main findings” do not sum up to 462 factors. Also, the author shall double check the numbers in the abstract.

Verification has been performed in the Excel file corresponding to the exhaustive list of chemical hazards. Finally, we have counted 26 chemicals in the category “others” like pointed out by the reviewer. Changes have been made in the Abstract (line 27), in Table 2 and in part 4.1 “Mains findings”.

4)    On page 12, there is a typo: "Aknowledgment" shall be “Acknowledgement”.

We changed the title : Aknowledgment “Acknowledgement”

5)    Suggest adding 462 category 1 chemicals in the supplemental files, even though Category 1 chemicals were reported in table 2. Adding CAS number would help to develop software/interview modules to overcome the difficulty for professionals to link such risks with environmental/occupational circumstances. 

We agree with the reviewer’s comment. The 462 chemicals from category 1 have been added in the supplemental files, with CAS numbers - when they were available in the lists used in our methodology.

Reviewer 2 Report

The manuscript titled “What environmental exposures to chemical hazards should be routinely investigated by physicians in perinatal preventive-care?” addresses important concerns regarding human exposure to environmental pollutants and raises awareness of exposure to these compounds particularly in critical windows of human development such as embryonic development. It is, in my opinion, extremely important to address occupational/environmental exposure to chemicals in a perinatal preventive-care perspective.

Furthermore, the manuscript is well written and the discussed studies are extremely relevant and solid from a scientific point of view.

However some issues must be addressed before publication:

Some minor errors are found such as in line 215 “Error! Reference…

In methods line 76, the drugs considered are only from occupational exposures? This should be clarified in order to avoid misunderstandings.

Considering the analysis performed, and the arguments presented in the discussion section, authors should provide not only the list of chemicals for category 1 but also Category 2 and 3.

It would also be interesting to discuss potential occupational/environmental scenarios where the exposure to the category 1 chemicals is the most possible considering the prevalence in consumer products, air etc…

Author Response

We thank the reviewer for his valuable comments and suggestions. We have made changes to the article in an effort to make it more intelligible and comprehensive. Hereinafter, we give our point-by-point answers to the concerns raised.

The reviewer made the four following comments:

1.     Some minor errors are found such as in line 215 “Error! Reference…”

We corrected this error as requested by the editor. “Error! Reference…” has been replaced by “Table 1”.

2.     In methods line 76, the drugs considered are only from occupational exposures? This should be clarified in order to avoid misunderstandings.

This is correct. We only addressed drugs from occupational exposure. We did not consider the medical treatment received by a person as “environmental exposure”.

In an effort to be more intelligible we reworded the sentence (line 76):

We also chose to exclude medical treatment and only consider drugs that could lead to occupational exposure.”

“We also chose to exclude medical treatment that cannot be considered as environmental exposure. Therefore, we included drugs when they could lead to occupational exposure in medical or research environments.”

3.     Considering the analysis performed, and the arguments presented in the discussion section, authors should provide not only the list of chemicals for category 1 but also Category 2 and 3.

The lists of hazards from Category 2 and 3 are presented in the supplemental file. Therefore, as requested by the first reviewer, we included the chemicals from Category 1 in the Table with their CAS number in order to facilitate the use of these lists in further works. Considering the significant size of this list (1,251 substances) it seemed appropriate to have the exhaustive list of hazards listed in the supplemental file.

4.     It would also be interesting to discuss potential occupational/environmental scenarios where the exposure to the category 1 chemicals is the most possible considering the prevalence in consumer products, air etc…

We completely agree with the importance to link the lists of chemical hazards with the environmental exposure circumstances to estimate the risk on reproductive health. We developed this idea in the first paragraph, part 4.4. “Implications and perspectives”. Unfortunately, it is difficult to deal similarly for hazards registered in category one in the current article considering the high number of circumstances that could be linked to each risk factor. This goes beyond the scope of this paper and should potentially be the subject of further work.